# Opinion: Influence of mean free path of air on atmospheric particle growth

Runlong Cai[1], Markku Kulmala[2]

[1]Shanghai Key Laboratory of Atmospheric Particle Pollution and Prevention (LAP[3]), Department of Environmental Science & Engineering, Fudan University, Shanghai, 200438, China

[2]Institute for Atmospheric and Earth System Research/Physics, Faculty of Science, University of Helsinki, Helsinki, 00014, Finland

*Correspondence to*: Markku Kulmala (markku.kulmala@helsinki.fi)

**Abstract**: Recent studies by Tsalikis et al. (2023, 2024) predicted that the mean free path of air ($\lambda_{air}$) could be significantly smaller than widely used values by a factor of ~2. Given the fundamental importance of $\lambda_{air}$, it has been a question whether an overestimation of $\lambda_{air}$ would have profound influences on a number of aerosol processes. Here we assume that the newly proposed value of $\lambda_{air}$ is accurate and examine its influences on our understanding of atmospheric aerosols. We show that for collisions-induced aerosol dynamics such as the condensation growth of atmospheric particles, the collision rate and hence the growth rate are determined by an effective mean free path for vapor and particle collision rather than $\lambda_{air}$. Similar to the cause of a smaller $\lambda_{air}$, the overlooked force field in pure elastic models may enhance vapor-particle collisions; however, this enhancement has been accounted for in previous studies. As a result, we find that the smaller $\lambda_{air}$ does not substantially influence particle collisions, i.e., it does not challenge the previous understandings of particle growth in the lower troposphere. Other potential influences on growth involving a high excess latent heat and the uncertainties in the sub-5 nm size range are also addressed.

The mean free path is a fundamental quantity in aerosol research. It is widely used to characterize the regime of particle motion in surrounding gases and is closely associated with particle dynamic processes such as deposition, diffusion, condensation, and coagulation. Recently, Tsalikis et al. (2023, 2024) investigated the mechanics and dynamics of collisions of air molecules (i.e., nitrogen and oxygen) and revisited the values of the mean free path of air ($\lambda_{air}$). Using a thoroughly validated atomistic molecular dynamics model, they proposed that the air molecules collide more frequently with each other compared to previous understandings. The force field, previously neglected in pure elastic models, was found to be the governing reason for the more frequent collisions. As a result, $\lambda_{air}$ was predicted to be significantly smaller than the widely used values. For example, the newly proposed value of $\lambda_{air}$ at 300 K and 1 atm is 38.5 nm, while the widely used value is 67.3 nm. However, the effects of this change in the value of $\lambda_{air}$ remain unclear. Here we assume that the newly proposed value of $\lambda_{air}$ is accurate and examine the influence of $\lambda_{air}$ on the dynamic processes of atmospheric aerosols.

We mainly focus on the collision rate between vapor molecules and aerosol particles, and subsequently, the growth rate of particles. This is motivated by the importance of growth to the atmospheric and climatic influences of airborne particles (Kulmala et al., 2022). If particles grew faster than previously expected, they could form more cloud condensation nuclei and provide more surface for atmospheric physicochemical processes. However, we find that despite the significant change in the value of $\lambda_{air}$, it has negligible influence on particle growth and relevant processes. That is, the framework of present-day aerosol dynamic theories, especially in typical atmospheric conditions, is not challenged by the uncertainties in the value of $\lambda_{air}$. We also note that the growth of small nanoparticles at the kinetic limit (e.g., < 3 nm) may be affected by the uncertainties associated with the force field of air molecules. The reasons are given in detail below.

The mean free path of air is the average distance an air molecule travels between its consecutive collisions with other air molecules. The relative size of a particle compared to $\lambda_{air}$ determines the behavior of air molecules surrounding it, which can be characterized by the Knudsen number ($Kn$),

$$Kn_{air} = \frac{2\lambda_{air}}{d_p} \tag{1}$$

where $d_p$ is the particle diameter and the subscript air emphasizes that Eq. 1 is for air molecules. The movement of particles is usually classified into three dynamic regimes according to the value of $Kn_{air}$, i.e., the kinetic regime (or free molecular regime) for $Kn_{air} \gg 1$, the continuum regime for $Kn_{air} \ll 1$, and the transition regime between the kinetic and continuum regimes for $Kn_{air} \sim 1$.

However, the use of the mean free path and its corresponding Knudsen number varies with the specific transfer process to particles. Fuchs and Sutugin (1971) proposed that the mean effective free path should be used and its formula differs for mass and heat transfer processes. For the collision between particles and condensable vapors, the mean effective free path characterizes the distance the vapor molecule travels before its velocity becomes independent of the initial velocity. Here we use a modified Fuchs-Sutugin expression for condensation reported in Lehtinen and Kulmala (2003), which has an improved accuracy for particles down to molecular sizes. The Knudsen number and mean free path are correspondingly defined as

$$\lambda = \frac{3(D_v + D_p)}{\sqrt{c_v{}^2 + c_p{}^2}} \tag{2}$$

$$Kn = \frac{2\lambda}{d_v + d_p} \tag{3}$$

where $D$ is the diffusion coefficient, $c$ is thermal velocity, and the subscripts v and p indicate vapor and particle, respectively. Different from Eq. 1, the $Kn$ in Eq. 3 is determined by the mean effective free path for vapor and particle collision, $\lambda$, instead of $\lambda_{air}$.

For the convenience of illustration, we first discuss the collision between vapors and small nanoparticles at the kinetic limit ($Kn \to \infty$). The collision coefficient can be expressed as

$$K_K = \frac{\pi}{4}\alpha(d_v + d_p)^2 \sqrt{\frac{8kT}{\pi m_v} + \frac{8kT}{\pi m_p}} \cdot E\left(\frac{A}{kT}, \infty\right) \tag{4}$$

where $K$ is the collision coefficient and the subscript K indicates the kinetic limit, α is the mass accommodation coefficient and it is taken to be 1 in this study, $m$ is mass, $E$ characterizes the enhancement of intermolecular forces on the collision coefficient compared to pure elastic collisions, A is the Hamaker constant, and $k$ is the Boltzmann constant, and $T$ is temperature. Here the relative thermal velocity between the colliding vapor and particle is explicitly expressed by their masses.

65        Equation 4 shows that the collision coefficient at the kinetic limit is governed by the collision cross-section and the relative thermal speed. While the relative thermal velocity is determined by the masses of vapor and particle, there may be uncertainties in the collision cross-section calculated using the vapor and particle diameters. Previous studies have shown that electrical mobility diameter inferred from measured electrical mobility using the Stokes-Millikan equation deviates from the mass diameter (Tammet 1995; Ehn et al., 2010; Larriba et al., 2011). Furthermore, as characterized by $E$ in Eq. 4, the attractive

component of the force field extends the time of colliding vapor and particle to stay close to each other. This is equivalent to increasing the collision cross-section for air molecules compared to pure elastic collisions. For instance, experimental evidence shows that van der Waals forces enhance the growth rate of newly formed particles by a factor of ~2 compared to growth by pure elastic collisions (Stolzenburg et al., 2020). This enhancement is also supported by molecular dynamic simulations for the collision between two sulfuric acid molecules (Halonen et al., 2019). Tsalikis et al. (2023, 2024) found similar influences

of the force field on the collision cross-section of air molecules, which is the governing reason for the smaller $\lambda_{air}$ predicted by molecular dynamic models compared to pure elastic models. We also note that the values of $\lambda_{air}$ and $\lambda$ are not used in computing the collision coefficient using Eq. 4; however, the influence of the force field of air molecules on the collision between vapors and particles remains to be further investigated.

       For collisions at the continuum and transition regimes, the collision coefficient between particles and vapors can be

expressed as

$$K = 2\pi\left(d_v + d_p\right)\left(D_v + D_p\right) \cdot \beta_m \cdot E\left(\frac{A}{kT}, Kn\right) \tag{5}$$

$$\beta_m = \frac{1 + Kn}{1 + 0.377Kn + \frac{4}{3\alpha}Kn(1 + Kn)} \tag{6}$$

where $\beta_m$ is the Fuchs-Sutugin semi-empirical factor. At the continuum limit ($Kn \rightarrow 0$), $\beta_m$ approaches 1 and Eq. 5 can be simplified as


$$K_C = 2\pi\left(d_v + d_p\right)\left(D_v + D_p\right) \cdot E\left(\frac{A}{kT}, 0\right) \tag{7}$$

where the subscript C indicates the continuum limit. At the kinetic limit ($Kn \rightarrow \infty$), it can be verified that Eq. 5 can be simplified to Eq. 4, i.e., the value of $K$ computed using Eq. 5 converges to the value of $K_K$ in Eq. 4 at the kinetic limit.

       Finally, the growth rate (d$d_p$/d$t$) of monodisperse particles due to vapor condensation can be obtained by multiplying the collision rate and the increment of particle size per collision, yielding


$$\frac{dd_p}{dt} = K(N_v - N_v^{eq})\Delta d_p \tag{8}$$

where $N_v$ is the concentration of vapor molecules, $N_v^{eq}$ is the equilibrium vapor concentration after considering the Kelvin effect and the effect of Raoult's law, and $\Delta d_p$ is the increment in particle diameter after adding a vapor molecule. For spherical particles, $\Delta d_p$ can be expressed as $\sqrt[3]{d_v^3 + d_p^3} - d_p$. Equations 2-8 can also be applied for coagulation processes by replacing the condensation vapors with particles, yet here we focus on condensation processes for the convenience of understanding.

For the convenience of understanding, we show the particle growth rate driven by the condensation of $10^7$ cm$^{-3}$ gaseous sulfuric acid in Fig. 1. The sulfuric acid is assumed to be non-volatile, and the Hamaker constant is taken to be $4.6 \times 10^{-20}$ J. According to Tsalikis et al. (2024), molecular dynamic simulation reproduced the values for the density, diffusion coefficient, and dynamic viscosity of air in experimental measurements or theoretical expressions well, and air velocity distribution was identical to classical expression. Equation 4 shows that the collision coefficient at the kinetic limit is determined by the sizes
and thermal velocities of vapors and particles, i.e., it is not affected by $\lambda_{air}$. Similarly, Eq. 7 shows that with given diffusion coefficients and thermal velocities, $\lambda_{air}$ does not influence the collision coefficient. Indeed, the diffusion coefficient of particles is calculated using $\lambda_{air}$; however, we will show that updating the value of $\lambda_{air}$ does not practically affect particle diffusion in the discussion below. For the transition regime ($0.1 < Kn < 10$), the formulas for the kinetic and continuum limits tend to overestimate the collision coefficient, and the Fuchs-Sutugin factor $\beta_m$ needs to be used. However, $\lambda_{air}$ is not used to calculate
the value of $\beta_m$ for vapors and particles with given diffusion coefficients and thermal velocities (Eqs. 2, 3, and 5). As a result, updating the value of $\lambda_{air}$ changes neither the particle growth rate nor the size range corresponding to kinetic, transition, and continuum regimes for vapor condensation (Fig. 1).

Figure 1


      The governing reason for the smaller $\lambda_{air}$ predicted by molecular dynamic models compared to pure elastic models is that the attractive component of the force field extends the time of colliding molecules to stay close to each other. This is equivalent to increasing the collision cross-section for air molecules compared to pure elastic collisions. As shown in Fig. 2, the measured particle growth rate of sub-10 nm particles exceeds the prediction by a pure elastic model ($A = 0$ and hence $E = 1$ in Eq. 5), in
which the condensing vapor (sulfuric acid) is assumed to be non-volatile and the growth rate is hence limited by the collision rate. This indicates a non-negligible collision enhancement by the force field between vapors and particles. With such a force field, the effective collision cross section is expected to be larger than that in the pure elastic model and the mean effective free path ($\lambda$) is correspondingly smaller than that predicted in Eq. 3. While noting that the influence of force field on $\lambda$ is expected to be size-dependent, we arbitrarily reduce the value of $\lambda$ by a constant factor of 2 and show a relatively good
consistency between the measured and predicted sub-10 nm growth rate.

Figure 2

However, previous studies have used a semi-empirical correction factor $E$ (see Eqs. 4, 5, and 7) to account for collision enhancement due to the force field between vapors and particles. With a Hamaker constant of $4.6 \times 10^{-20}$ J, the growth rate predicted using Eq. 4 can reproduce the measured data without adjusting the value of $\lambda$ (Fig. S1). That is, the influence of the force field on vapor condensation has been accounted for in present-day particle growth studies, though this is not achieved by explicitly updating the value of $\lambda$. Previous studies (e.g., Stolzenburg et al., 2020) have also discussed the uncertainty in $E$, which is associated with the uncertainty in $\lambda$ for growth rate calculation. Whichever method is implemented for this enhancement, it is important to avoid double-counting: if one uses the van-der-Waals correction factor $E$ for the force field, one should not adjust $\lambda$ and vice versa.

The diffusion coefficient of particles is a function of the $\lambda_{\text{air}}$. The diffusion coefficient $D_{\text{p}}$ can be expressed as

$$D_{\text{p}} = \frac{kTC_{\text{s}}}{3\pi\mu d_{\text{p}}} \tag{9}$$

$$C_{\text{s}} = 1 + Kn_{\text{air}}\left(a + b \cdot e^{-c/Kn_{\text{air}}}\right) \tag{10}$$

where $C_{\text{s}}$ is the Cunningham slip correction factor, $\mu$ is the viscosity of air, and $a$, $b$, and $c$ are fitting parameters for the Knudsen-Weber formula for $C_{\text{s}}$.

Equations 9-10 indicate that the value of $\lambda_{\text{air}}$ influences particle diffusion by influencing $C_{\text{s}}$. However, these influences do not apply to the update of $\lambda_{\text{air}}$ proposed by Tsalikis et al. (2023, 2024). This is because the values of the fitting parameters in Eq. 10 were obtained as the best fit to experimental results with a predetermined value of $\lambda_{\text{air}}$. Updating the value of $\lambda_{\text{air}}$ corresponds to an update in the values of fitting parameters rather than the values of $C_{\text{s}}$. For instance, the values of $a$, $b$, and $c$ reported in Allen and Raabe (1985) for spherical solid particles for $\lambda_{\text{air}} = 67.3$ nm at sea level 23°C are 1.142, 0.558, and 0.999, respectively. If $\lambda_{\text{air}}$ is updated to 38.5 nm at the same temperature and pressure, $a$, $b$, and $c$ should be correspondingly updated to 1.996, 0.975, and 1.746, respectively, to be consistent with experimental results from the Millikan oil-drop experiments. For the same reason, updating the value of $\lambda_{\text{air}}$ by accounting for the force field for collisions is not expected to directly affect other processes relevant to the drag force of air molecules on particles at atmospheric conditions, e.g., coagulation, agglomeration, and electrical motion, though indirect influences may exist. It is also worth clarifying that the diffusion coefficients of vapor and particle in Eq. 5 cancel off with that in $\beta_{\text{m}}$ at the kinetic limit (see also Eq. 4). Consequently, the uncertainty of Eq. 9 for small nanoparticles (e.g., <3 nm) does not propagate into the value of $K$.

For the rapid condensation of vapor molecules with high latent heat, the heat transfer from growing aerosol particles needs to be considered (e.g., Yang et al., 2019). In these situations, the temperature at the aerosol surface can be significantly higher than the ambient air temperature. One typical example in the atmosphere is cloud droplet formation, during which supersaturated water vapor condenses on cloud condensation nuclei (ca 100 nm) within a few minutes to form cloud droplets (ca 10000 nm) (e.g., Kulmala et al., 1993). Different from the $Kn$ for vapor condensation in Eq. 3, the Knudsen number for heat transfer should be $Kn_{\text{air}}$, which is defined based on $\lambda_{\text{air}}$. However, since the heat and mass transfer for cloud droplet formation occurs mainly in the continuum regime, the droplet growth rate has a negligible dependence on $\lambda_{\text{air}}$.

In summary, we find that the smaller $\lambda_{air}$ predicted by atomistic molecular dynamics than widely used values predicted by pure elastic collision models does not substantially influence particle growth by condensation and coagulation. This is because the collision of particles is determined by the mean effective free path of condensable vapors rather than $\lambda_{air}$, and the formula for the slip correction factor is fitted to experimental results with a predetermined value of $\lambda_{air}$. The influences of the force field of air molecules on the collision between vapors and particles may need further investigation. For rapid growth processes involving a high excess latent heat, a smaller $\lambda_{air}$ affects growth by increasing heat transfer flux. However, this is not the case for atmospheric nanoparticle growth in the lower troposphere, and it has only a minor influence on cloud droplet formation.

## Code availability

The code for the figure can be provided upon request from the corresponding author.

## Author contributions

MK conceptualized the research; RC performed model simulations; MK and RC analyzed the results; RC wrote the manuscript with inputs from MK.

## Competing interests

One of the authors is a member of the editorial board of Aerosol Research. The peer-review process was guided by an independent editor, and the authors also have no other competing interests to declare.

## Acknowledgements

We thank Sotiris E. Pratsinis and Claudia Mohr for their insightful discussions.

## Financial support

This research has been supported by the National Natural Science Foundation of China (grant no. 22406024), ACCC Flagship funded by the Academy of Finland grant number 337549 (UH) and 337552 (FMI), Academy professorship funded by the Academy of Finland (grant no. 302958), Academy of Finland projects no. 1325656, 311932,334792, 316114, 325647, 325681, 347782, 324259, the Strategic Research Council (SRC) at the Academy of Finland (#352431), "Quantifying carbon sink, CarbonSink+ and their interaction with air quality" INAR project funded by Jane and Aatos Erkko Foundation, "Gigacity" project funded by Wihuri foundation, grant VN/28414/2021 by the Ministry of Agriculture and Forestry of Finland, European Union (Grant no. 101059888, Climb-Forest and Grant no. 871128, eLTER PLUS), European Research Council (ERC) project ATM-GTP Contract No. 742206, and European Union via Non-CO2 Forcers and their Climate, Weather, Air Quality and Health Impacts (FOCI).

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

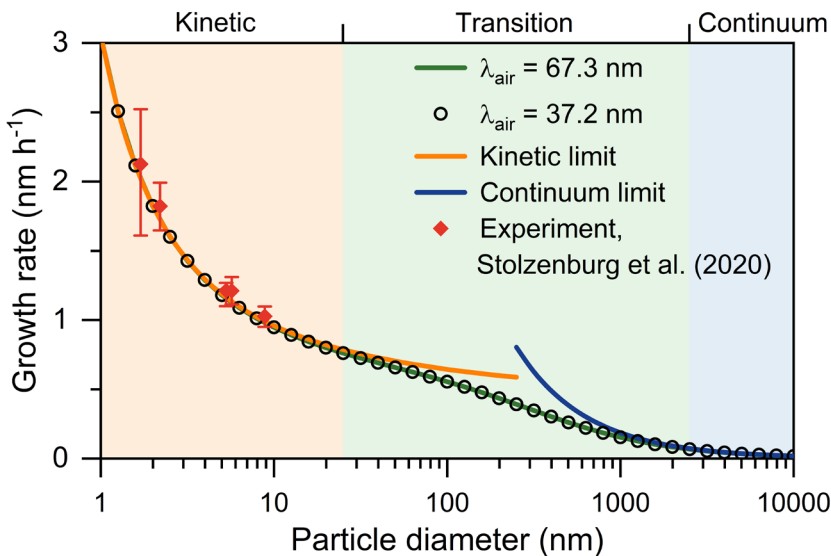

**Figure 1: Particle growth rate by the condensation of sulfuric acid.** The sulfuric acid concentration is $10^7$ cm$^{-3}$. Lines and open markers are theoretically predicted results using Eqs. 4-8. The shaded areas indicate the dynamic regimes for collision, i.e., the kinetic regime ($Kn \geq 10$), the transition regime ($0.1 < Kn < 10$), and the continuum regime ($Kn \leq 0.1$). Closed markers
indicate the experimentally determined growth rate from chamber experiments reported by Stolzenburg et al. (2020).

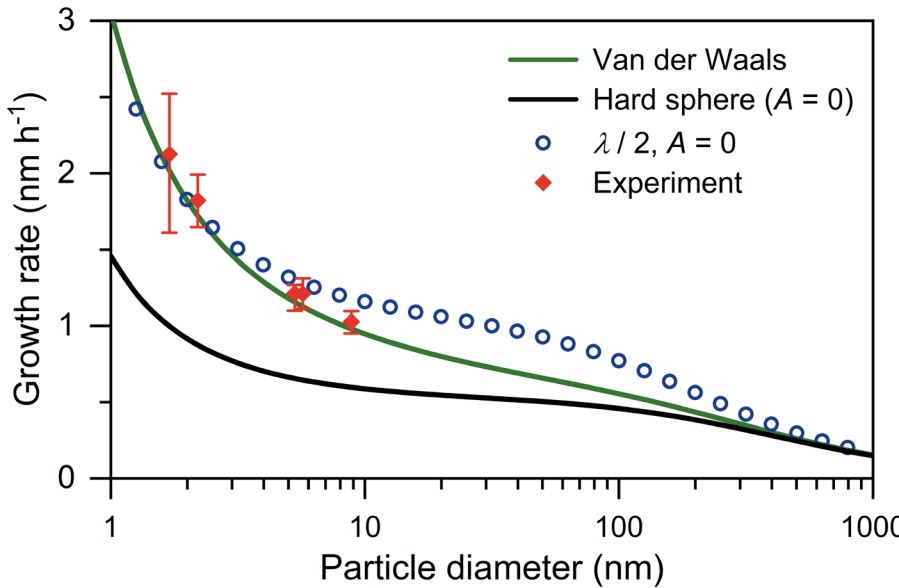

**Figure 2: Influence of the force field on particle growth rate.** The growth rate is calculated for the condensation of non-volatile sulfuric acid with a concentration of $10^7$ cm$^{-3}$. The green line (van der Waals) indicates the results calculated using Eq. 5 with a Hamaker constant ($A$) of $4.6\times10^{-20}$ J. The black line (hard sphere) is calculated using a pure elastic model in which particles and vapors are taken as hard spheres, i.e., by setting the value of A in Eq. 5 to 0. Open markers indicate the growth rates with $\lambda$ values hypothetically divided by a factor of 2. Closed markers are from Stolzenburg et al. (2020).