# Peer review of "Opinion: Influence of mean free path of air on atmospheric particle growth"

_Aerosol Research, 2025_

## Author Response (AR1)

**Responses to Reviewers' Comments on Manuscript ID AR-2025-8**

(Opinion: Influence of mean free path of air on atmospheric particle growth)

We thank the two anonymous reviewers for their comments and Prof. Ben Murray for handling this manuscript. We have revised the manuscript and moved the materials from the supplement to the main text, mainly to address the comments on the vapor mean free path. We believe that the influences of the force field on the vapor mean free path and particle growth rate are clarified in the revised manuscript. All changes made in the manuscript are highlighted in the revised manuscript and shown as shown as "quoted underlined texts" in the responses. The reviewers' comments are shown as sans-serif blue text and our responses are shown as serif black text.

**Reviewer #1:**

The paper points out that the growth rate of aerosol particles is little affected by a recently proposed uncertainty (Tsalikis et al (2023), (2024)) in the mean free path of air molecules. This is because the relevant molecular path length affecting the rate of growth is the mean free path of vapor molecules prior to collisions with an aerosol particle rather than the mean free path of air. Numerous equations for the aerosol growth rate for different regimes of particle size are reviewed to demonstrate this.

The more interesting discussion in this paper lies in the Supplement, where an uncertainty in the vapor mean free path of a similar degree to that proposed for air is shown to make a substantial difference to the particle growth rate. I do wonder why the authors claim that the vapor mean free path is not uncertain when recent studies suggest there is uncertainty in the mean free path for air molecules. I recommend that the authors bring the material in the Supplement into the main body of the paper and comment further on this matter.

**Response:** We thank the reviewer for this very constructive comment. To better clarify the uncertainty in the vapor mean free path  $\lambda$  and its influences on the mean free path, we revised the main text, including removing a potentially misleading statement that "we do not expect this (hypothetically changing the value of  $\lambda$ ) to happen", rewrote the supplement, and moved the materials in the supplement to the main text.

Briefly, we clarify in the revised manuscript that  $\lambda$  is expected to be influenced by the force field; however, as such an influence on the growth rate has been accounted for by the collision enhancement factor *E*, there is no necessity to explicitly change the value of  $\lambda$  to avoid double counting. Accordingly, the uncertainty in  $\lambda$  for growth rate calculation is reflected by the uncertainty in the collision enhancement factor *E*. We think that the revised contents, together with existing discussions (e.g., in lines 65-78) can clarify this matter. The revised text is given at the end of the responses.

**Reviewer #2:**

This opinion piece of Cai & Kulmala tests whether recent revisions to the mean free path of air alter the growth rates of aerosol particles. They present a thorough and very well written theoretical overview, before a handful of calculations showing that there is no notable difference. I echo the other reviewer that the discussion in the supplement is interesting, but I am ambivalent about whether it belongs in the main text. I have no specific comments. This work is highly useful and I enjoyed reading it, and therefore recommend it for publication.

**Response:** We thank the kind comment from the reviewer. After consideration, we choose to move the materials on  $\lambda$  from the supplement to the main text. We revised the relevant sentences in the main text and also rewrote the materials in the original supplement for better clarification. The revised main text is given at the end of the responses.

Briefly, there is expected to be a notable difference in  $\lambda$ ; however, instead of changing the values of  $\lambda$ , present-day studies often use a collision enhancement factor to account for the enhanced growth rate. We clarified in the revised manuscript that it is important to avoid double counting whichever method to use for the growth rate correction.

**Revised main text**

**Main text (Lines 112-131)**

"This is equivalent to increasing the collision cross-section for air molecules compared to pure elastic collisions. As shown in Fig. 2, the measured particle growth rate of sub-10 nm particles exceeds the prediction by a pure elastic model (A = 0 and hence E = 1 in Eq. 5), in which the condensing vapor (sulfuric acid) is assumed to be non-volatile and the growth rate is hence limited by the collision rate. This indicates a non-negligible collision enhancement by the force field between vapors and particles. With such a force field, the effective collision cross section is expected to be larger than that in the pure elastic model and the mean effective free path ( $\lambda$ ) is correspondingly smaller than that predicted in Eq. 3. While noting that the influence of force field on  $\lambda$  is expected to be size-dependent, we arbitrarily reduce the value of  $\lambda$  by a constant factor of 2 and show a relatively good consistency between the measured and predicted sub-10 nm growth rate.

However, previous studies have used a semi-empirical correction factor *E* (see Eqs. 4, 5, and 7) to account for collision enhancement due to the force field between vapors and particles. With a Hamaker constant of  $4.6 \times 10^{-20}$  J, the growth rate predicted using Eq. 4 can reproduce the measured data without adjusting the value of  $\lambda$  (Fig. S1). That is, the influence of the force field on vapor condensation has been accounted for in present-day particle growth studies, though this is not achieved by explicitly updating the value of  $\lambda$ . Previous studies (e.g., Stolzenburg et al., 2020) have also discussed the uncertainty in *E*, which is associated with the uncertainty in  $\lambda$  for growth rate calculation. Whichever method is implemented for this enhancement, it is important to avoid double-counting: if one uses the van-der-Waals correction factor *E* for the force field, one should not adjust  $\lambda$  and vice versa."

**Figure 2: Influence of the force field on particle growth rate.** The growth rate is calculated for the condensation of non-volatile sulfuric acid with a concentration of  $10^7$  cm-3. The green line (van der Waals) indicates the results calculated using Eq. 5 with a Hamaker constant (*A*) of  $4.6 \times 10$ -20 J. The black line (hard sphere) is calculated using a pure elastic model in which particles and vapors are taken as hard spheres, i.e., by setting the value of A in Eq. 5 to 0. Open markers indicate the growth rates with  $\lambda$  values hypothetically divided by a factor of 2. Closed markers are from Stolzenburg et al. (2020)."

---

## Author Response (AR2)

**Responses to Editor's Comments on Manuscript ID AR-2025-8**

(Opinion: Influence of mean free path of air on atmospheric particle growth)

Editor's comment

In this Opinion paper, Cai and Kulmala comment on the implications of recent updates on the mean free path of air molecules on aerosol growth. They conclude that current schemes account for these effects, but also note that these effects must not be double counted in future studies. The authors have done a good job at responding to the referees' largely positive comments.

I only have one technical correction. I would like to be a little more specific in the competing interests statement, I suggest:

One of the co-authors is a member of the editorial board of Aerosol Research. The peer-review process was guided by an independent editor, and the authors also have no other competing interests to declare.

Response: We thank the editor again for handling the manuscript and giving very helpful comments. The competing interests statement has been revised as suggested.